# Cu@Pd/C with Controllable Pd Dispersion as a Highly Efficient Catalyst for Hydrogen Evolution from Ammonia Borane

**DOI:** 10.3390/nano10091850

**Published:** 2020-09-16

**Authors:** Yanliang Yang, Ying Duan, Dongsheng Deng, Dongmi Li, Dong Sui, Xiaohan Gao

**Affiliations:** 1Henan Key Laboratory of Function-Oriented Porous Materials, College of Chemistry and Chemical Engineering, Luoyang Normal University, Luoyang 471934, China; dengdongsheng168@sina.com (D.D.); lidongmi223@126.com (D.L.); suidonghy@mail.nankai.edu.cn (D.S.); 2College of Food and Drug, Luoyang Normal University, Luoyang 471934, China; duanying@mail.ustc.edu.cn; 3School of Chemistry and Material Science, College of Chemistry, Chemical Engineering and Environmental Engineering, Liaoning Shihua University, Fushun 113001, China

**Keywords:** Pd dispersion, galvanic reduction, partial oxidation, ammonia borane

## Abstract

A series of Cu@Pd/C with different Pd contents was prepared using the galvanic reduction method to disperse Pd on the surface of Cu nanoparticles on Cu/C. The dispersion of Pd was regulated by the Cu(I) on the surface, which was introduced by pulse oxidation. The Cu_2_O did not react during the galvanic reduction process and restricted the Pd atoms to a specific area. The pulse oxidation method was demonstrated to be an effective process to control the oxidization degree of Cu on Cu/C and then to govern the dispersion of Pd. The catalysts were characterized by transmission electron microscopy (TEM), high-resolution transmission electron microscope (HRTEM), high angular annular dark field scanning TEM (HAADF-STEM), energy-dispersive spectroscopy (EDS) mapping, X-ray diffraction (XRD), X-ray photoelectron spectroscopy (XPS), auger electron spectroscopy (AES), and inductively coupled plasma optical emission spectrometer (ICP-OES), which were used to catalyze the hydrogen evolution from ammonia borane. The Cu@Pd/C had much higher activity than the PdCu/C, which was prepared by the impregnation method. The TOF increased as the Cu_2_O in Cu/C used for the preparation of Cu@Pd/C increased, and the maximum TOF was 465 mol_H2_ min^−1^ mol_Pd_^−1^ at 298 K on Cu@Pd_0.5_/C-640 (0.5 wt % of Pd, 640 mL of air was pulsed during the preparation of Cu/C-640). The activity could be maintained in five continuous processes, showing the strong stability of the catalysts.

## 1. Introduction

Safe and convenient hydrogen storage is one of the key problems that have emerged in the application of hydrogen as clean energy. A lot of effort has been devoted to developing hydrogen storage materials, including physical and chemical storage systems, with excellent performance. Ammonia borane (AB), a stable solid in normal conditions with a hydrogen content of 19.6 wt %, was thought to be a potential chemical hydrogen storage material and has gained the attention of many scientists [1,2,3,4,5,6,7,8,9,10]. The AB could release 3 mol H_2_ per mole AB at mild conditions in the presence of an appropriate catalyst.

A lot of catalysts were reported to be effective for the hydrogen evolution reaction from AB. Noble metals usually had very high activity for the hydrolysis of AB [11,12,13,14,15,16,17,18,19,20,21,22,23,24,25,26,27,28,29,30,31,32,33,34,35,36,37,38,39,40]. However, as noble metals are expensive, the full utilization of the noble metal atoms is a significant issue in the production of noble metal-based catalysts. The non-noble metal-based catalysts were also used to catalyze the hydrolysis of AB, and the well-designed catalytic system had attractive activity for this reaction [41,42,43,44,45,46,47,48,49,50,51,52,53,54,55]. However, most of the non-noble metal-based catalysts had lower catalytic activity and less stability compared to the noble metals. Thus, attaining stable catalysts with highly activity and low cost was the aspirational aim for the hydrolysis of AB.

To reduce the consumption of noble metals, the addition of another non-noble metal was a useful option. The electronic structures of the noble metal catalysts could be modulated by the non-noble metal to obtain a more efficient bimetallic catalyst [30,56,57,58,59,60,61,62,63,64]. During the preparation of the bimetallic catalyst, burying the noble metal atoms in the metal nanoparticles should be avoided to get full utilization of the noble metals. The galvanic reduction method was an efficient approach to avert the embedment of noble metals, as the reduction reaction only occurred on the surface of non-noble metal. For example, Chen et al. achieved an atomically dispersion of Pt on the surface of Ni particles through the galvanic reduction method [62]. Dong et al. coated Co nanoparticles by Pt through the spontaneous displacement reaction of Co by Pt. As a result, the Pt covered the surface of Co nanoparticles only [57]. Both of the catalysts showed excellent catalytic activity on the hydrolytic dehydrogenation of ammonia borane.

The galvanic reduction method was an effective process to get a surface noble metal-rich catalyst [65,66,67,68,69,70,71,72]. However, when the pure metallic non-noble metal nanoparticles were used for the galvanic reduction process, the noble metal atoms tended to aggregate into large particles on the surface of the non-noble metal as on other supports. As a result, how to control the distribution of noble metals on the surface of non-noble metals needs further consideration. In this manuscript, based on our previous work on the preparation of Pd catalysts [73,74,75,76,77,78], we synthesized Cu@Pd/C with controllable Pd dispersion through the galvanic reduction method. The dispersion of Pd atoms on the surface of Cu nanoparticles was regulated by partial oxidation of the surface of Cu nanoparticles. The catalysts showed an initial turnover frequency (TOF) of 465 mol_H__2_ min^−1^ mol_Pd_^−1^ at 298 K for the hydrolysis of AB, which was among the highest values in the literatures.

The preparation of Cu@Pd/C catalysts is illustrated in Figure 1. Firstly, Cu/C was synthesized by carbothermal reduction in N_2_. Then, the surface of the Cu was partially oxidized to Cu(I) by the pulse injection of a limited amount of air at 473 K. The Pd was introduced to the surface of Cu particles by the displacement reaction of Cu(0) by Pd(II). As the displacement reaction was only conducted on Cu(0), the Pd(0) was confined to a limited area surrounded by Cu(I). As a result, the aggregation of Pd atoms to form larger particles was prevented by the surrounding Cu(I) species.

## 2. Materials and Methods

### 2.1. Preparation of Catalysts

#### 2.1.1. Cu/C

The Cu/C was prepared using the impregnation method. Typically, Cu(NO_3_)_2_·3H_2_O (1.20 g) was dissolved in water (5.65 g) to form a homogeneous transparent solution. Then, AC (5.00 g, 100 mesh) was added into the solution and stirred into paste. After being dried at 373 K overnight, the solid was transferred into a tubular furnace and calcined at 773 K for 2 h in N_2_ atmosphere to afford the Cu/C.

#### 2.1.2. Cu/C-x

The Cu/C (1.00 g) was heated to 473 K in a tubular furnace under N_2_ atmosphere. Then, the air was pulse-injected into the furnace at a rate of 40 mL min^−1^. Eighty mL of air was introduced for each pulse, and the interval for each pulse was 1 min. After all the air needed was introduced, the sample was cooled to room temperature quickly in N_2_. The sample was denoted as Cu/C-x, where x referred to the amount of air pulsed.

#### 2.1.3. Cu@Pd_y_/C-x

The PdCl_2_ dissolved in dilute HCl (Pd, 0.5 wt %) was added dropwise into a suspension of Cu/C-x (0.50 g) in water (40 g) under N_2_ atmosphere at room temperature. The suspension was stirred for 4 h and washed with water three times and then with ethanol once. The solid was dried under vacuum at 323 K to afford the Cu@Pd_y_/C-x where y was the mass percent of Pd.

#### 2.1.4. Pd_y_Cu/C

The Pd_y_Cu/C was prepared using the impregnation method: the same method used for Cu/C, except instead of the solution of Cu(NO_3_)_2_·3H_2_O a solution of PdCl_2_ and Cu(NO_3_)_2_·3H_2_O was used. The following process was the same as the preparation of Cu/C.

### 2.2. Characterization

The high-resolution transmission electron microscope (HRTEM) and transmission electron microscopy (TEM) mappings were collected on a Tecnai G2 F20 (Hillsboro, OR, USA). The energy-dispersive spectroscopy (EDS) mappings were characterized by a JEM2100F (Tokyo, Japan) equipped with an EDS detector. The high angular annular dark field scanning TEM (HAADF-STEM) images were taken on a FEI Themis Z microscope (Hillsboro, OR, USA). The X-ray diffraction (XRD) patterns were obtained on a Rigaku D/Max 2500/PC powder diffractometer (Tokyo, Japan) with Cu Kα radiation (λ = 0.15418 nm) at 40 kV. The scanning rate was 5° min^−1^. Inductively coupled plasma optical emission spectrometer (ICP-OES) was conducted on Agilent 5110 ICP-OES equipment (Santa Clara, CA, USA). The samples were calcined in air at 673 K before dissolution. X-ray photoelectron spectroscopy (XPS) and auger electron spectroscopy (AES) measurements were conducted on a Thermo Escalab 250Xi spectrometer (Waltham, MA, USA).

### 2.3. Catalytic Hydrolysis of AB

The catalytic hydrolysis of AB was conducted on a homemade reaction apparatus for monitoring hydrogen generation. Firstly, H_2_O (9.5 mL) and catalyst (Pd, 0.45 mg) was loaded into the jacket reactor connected to constant temperature water. The reactor was charged and purged with N_2_ for 3 times and stirred for 30 min at the desired temperature to reach thermal equilibrium. Then, the reaction was started by the introduction of a solution of AB (0.5 mL, 1 mmol) into the reactor. The volume of H_2_ was monitored by introducing the gas to a calibrated glass tube at scale. The tube was surrounded by constant water (298 K), and the bottom of the tube was connected to a burette opening to the atmosphere. The liquid level of the tube and the burette was controlled to the same height to ensure the pressure in the tube was maintained at 1 atmospheric pressure.

## 3. Results and Discussion

### 3.1. Charcaterization

The EDS mapping of Cu/C-320 is shown in Figure 2a–e. The Cu, C, and O were detected in the sample. The distribution of O did not always follow the distribution of Cu. In other words, the signal of O and Cu did not always appear in the same areas. This phenomenon can be explained in this way. When an area had the signal of Cu without O, it should be the signal for Cu(0). The simultaneous appearance of Cu and O was mostly as the oxidized Cu. So, both Cu(0) and oxidized Cu were contained in Cu/C-320 revealed by the EDS mapping. This result was confirmed by the XRD and HRTEM characterization. The diffraction peaks corresponding to both Cu and Cu_2_O appeared in the XRD pattern of Cu/C-320 (Figure 3a). The HRTEM image of Cu/C-320 was exhibited in Figure 2f. The crystalline domains corresponding to both Cu (d_111_ = 0.210 nm) and Cu_2_O (d_110_ = 0.305 nm) could be observed in the image. Based on the above characterization, the Cu/C-320 should possess a surface composed of Cu and Cu_2_O after the pulse injection of air.

The Cu@Pd/C-320 was prepared by the displacement of surface Cu(0) by Pd(II) in N_2_. The theoretical contents of Pd were varied from 0.5 wt % to 2.0 wt %. The XRD patterns of Cu@Pd/C-320 are shown in Figure 3a. The diffraction peaks centered at 36.4°, 42.3°, 61.3°, and 73.5° were attributed to the diffraction of Cu_2_O, while the peaks at 43.3°, 50.4°, and 74.1° were the diffractive peaks of metallic Cu. The diffraction peaks corresponding to Cu(0) decreased as the Pd content increased, implying the replacement of Cu(0) by Pd(II). The diffraction peaks of Cu_2_O had broadened slightly while maintaining intensity as the Pd content increased, showing that the Cu_2_O was steady during the replacement process. No peaks corresponding to Pd were detected, which should be caused by the high dispersion of Pd. The HRTEM also showed no crystalline domains of Pd for the Cu@Pd_0.5_/C-320 and Cu@Pd_2.0_/C-320 samples (Figure 3b). As the Cu(0) was surrounded by Cu(I), only very little Pd(II) could be reduced at a special area. After the displacement, the Pd(0) had little chance to get together, as they were cut apart by the Cu(I). The EDS mappings of Cu@Pd/C-320 are displayed in Figure 3c–g and Appendix A–S3. The Pd had uniform distribution in all four samples with different Pd content. After careful analysis of the element distribution in the EDS mappings, it could be seen that the Pd mostly embedded on the surface of Cu nanoparticles. In the area determined in the blue box in Cu@Pd_0.5_/C-320 (Figure 3c), the Pd was surrounded by O, which should have originated from the Cu_2_O. This was clear proof for the assumption that the Cu_2_O divided the surface of Cu(0) and surrounded the Pd to prevent the aggregation of Pd atoms. The distribution of Pd on Cu was characterized by HAADF-STEM, and the result is shown in Figure 3h–i. Both single Pd atoms and two-dimensional clusters can be seen in the HAADF-STEM images. This showed that the strategy was effective for the preparation of Pd catalysts with high Pd dispersion.

To make a comparison, the PdCu/C catalysts were prepared through the impregnation method with the same theoretical contents of Pd as Cu@Pd/C-320. The XPS spectra of Cu@Pd/C-320 and PdCu/C-320 are displayed in Figure 4. The binding energy at 336.4 eV and 341.9 eV were assigned to pd 3d_5/2_ and pd 3d_3/2_ [79,80]. The Cu/C-320 had no signal of Pd. The intensity of the Pd signal increased as the Pd content increased for both Cu@Pd/C-320 and PdCu/C. However, the Cu@Pd/C-320 had higher intensity peaks at 336.4 eV and 341.9 eV than that of PdCu/C at the same content of Pd. As only the surface photoelectrons could escape from the sample, the XPS characterization showed the surface composition of samples. The higher peaks intensity showed that the Cu@Pd/C-320 possessed a Pd-rich surface comparing to PdCu/C.

### 3.2. Hydrogen Evolution from AB

The hydrogen evolution reaction was conducted on a homemade device, as shown in Appendix A. Both the Cu@Pd/C and PdCu/C were employed as the catalysts for the hydrolysis of AB at 298 K, and the results are shown in Figure 5a. The activated carbon had no activity for the hydrolysis of AB, and the commercial Pd/C had a low hydrogen evolution rate (3.5 mol_H2_ min^−1^ mol_Pd_^−1^) in the reaction conditions. The Cu@Pd_0.5_/C-320 achieved a TOF of 306 mol_H2_ min^−1^ mol_Pd_^−1^ at 298 K, which was twice as much as that of Pd_0.5_Cu/C (149 mol_H2_ min^−1^ mol_Pd_^−1^) at the same conditions. As the content of Pd increased to 2.0%, the TOF of Pd_2.0_Cu/C decreased sharply to 53 mol_H2_ min^−1^ mol_Pd_^−1^, while that of Cu@Pd_2.0_/C-320 decreased very slowly to 256 mol_H2_ min^−1^ mol_Pd_^−1^. The difference in the activity between PdCu/C and Cu@Pd/C-320 should be caused by the different Pd distribution. As the hydrolysis of AB happened on the surface of the catalysts, the activity mostly depended on the surface Pd content. The Pd was mainly distributed on the surface of Cu nanoparticles in Cu@Pd/C-320, while some of the Pd atoms may be buried in the Cu nanoparticles for PdCu/C. As a result, the Cu@Pd/C-320 had higher TOF than that of PdCu/C. As the content of Pd increased, more Pd atoms would be buried in the nanoparticles for PdCu/C, leading to the sharp decrease in TOF. On the other hand, the galvanic reduction method ensured that most of the Pd atoms distributed on the surface for Cu@Pd/C-320, disregarding the content of Pd. As a result, the activity decreased a little as the Pd content increased. Similar trends were obtained as the reaction temperature increased from 298 to 313 K. The hydrolysis of AB catalyzed by Cu@Pd/C-320 and PdCu/C were conducted at 298, 303, 318, and 313 K respectively (Figure 5a and Appendix A). The TOF on Cu@Pd/C-320 was much higher than the TOF on PdCu/C with the same Pd content at different temperatures (Figure 5b,c). The TOF on Cu@Pd/C-320 remained steady, while that on PdCu/C decreased sharply as the contents of Pd at a certain temperature increased. The activation energy (*Ea*) was calculated based on the Arenius formula (Figure 5b,c). It was found that both the Cu@Pd/C-320 and PdCu/C had similar *Ea* values (around 60 kJ mol^−1^). This implied that the catalysts might have similar active site considering the similar composition of the catalysts, and the difference in TOF should not be caused by the change in *Ea*.

### 3.3. Effect of Cu(I)

To illustrate the role of Cu(I) during the preparation of catalysts, Cu/C with different Cu_2_O content were prepared and used for the synthesis of Cu@Pd/C. The oxidization degree of Cu on Cu/C was controlled by changing the amount of air injected. The XRD patterns of Cu/C are shown in Figure 6a. After the carbothermal reduction in N_2_, diffraction peaks corresponding to metallic Cu were observed in Cu/C, and no diffraction peaks of Cu_2_O were found in the XRD pattern. Then, the air was injected under N_2_ atmosphere at 498 K to partially oxidize the Cu nanoparticles on Cu/C-0. After the injection of 160 mL of air, diffraction peaks ascribed to Cu_2_O emerged in the XRD pattern of Cu/C-160. As the amount of air injected increased, the diffraction peaks corresponding to Cu_2_O increased, while those corresponding to Cu decreased. Two tiny diffraction peaks (35.5° and 38.7°) ascribed to CuO could be observed on Cu/C-480. When 640 mL of air was injected, diffraction peaks of metallic Cu disappeared, and the peaks of CuO enlarged a little. It is obvious that the Cu was controllably oxidized to Cu_2_O by the injected air, and the large amount of air led to the presence of CuO. The oxidation states of the surface Cu were further characterized by XPS (Figure 6b) and AES (Figure 6c). The satellite peak of CuO at 943.4 eV in the XPS spectra was not obvious in all the Cu/C samples except for Cu/C-640. The Cu/C-640 sample showed a tiny but obvious CuO satellite peak in the XPS spectra [81]. This showed that the CuO was not the main species on the surface of Cu nanoparticles. However, the peak areas of CuO at 933.7 eV did increase as the amount of air increased. As the Cu_2_O and Cu had similar binding energy in the XPS spectra, the AES spectra was employed to distinguish the Cu_2_O and Cu. The binding energy at 568.5 eV was considered to be the signal of CuO and Cu [81,82]. The low content of CuO on the surface was revealed by the XPS results, which showed that this 568.5 eV peak was mostly caused by the metallic Cu. The peak centered at 570.0 eV was ascribed to the Cu_2_O [81,82]. It could be seen that there was a small portion of Cu_2_O in Cu/C-0 after the carbothermal reduction in N_2_. This could be caused by the short exposure to air during the preparation process. As the amount of air injected increased, the peak ascribed to Cu_2_O increased, while the peak ascribed to Cu decreased. The results indicated that more and more metallic Cu was oxidized on the surface as more air was introduced. The content of Cu species on the surface could be easily regulated by the control of air injected.

The Cu/C with different Cu_2_O content was used for the preparation of Cu@Pd/C by the galvanic reduction method. The hydrolysis of AB was conducted to evaluate the Cu@Pd/C prepared from Cu/C with different Cu_2_O content. The results are shown in Figure 7. The TOF on Cu@Pd/C-0 was 180 mol_H__2_ min^−1^ mol_Pd_^−1^, which was much higher than the TOF on commercial Pd/C and Pd_0.5_Cu/C prepared by the impregnation method, showing the superiority of the galvanic reduction method. The activity could be further improved by the partial oxidation of the surface of Cu nanoparticles on Cu/C before the galvanic reduction process. The TOF increased as the surface oxidation of Cu was enhanced. The reason for this increase could be explicated as follows. The surface of Cu nanoparticles was divided by Cu(I) into discrete pieces of metallic Cu, which acted as a reducing agent for Pd(II). The Cu(I) did not react with the Pd(II) during the galvanic reduction process. As a result, the reduced Pd(0) was confined to the original position of Cu(0), and the aggregation of Pd(0) was restricted by the Cu(I) during the preparation process. The best TOF value (465 mol_H__2_ min^−1^ mol_Pd_^−1^) was acquired on Cu@Pd/C-640 owing to its highest content of Cu_2_O. Table 1 summarizes the TOF values for the reported palladium catalysts in the hydrolysis of AB. It can be seen that the TOF values on Cu@Pd/C were among the highest TOF values reported in the literature.

### 3.4. Catalyst Stability

The stability of the catalysts was studied using Cu@Pd_0.5_/C-320 as catalysts at 298 K. After each reaction, another 1 mmol of AB was added into the reactor to record the hydrogen released from AB vs. time. The results are shown in Figure 8 and Appendix A. The catalytic performance was well maintained in five continuous processes. The used catalyst Cu@Pd_0.5_/C-320-R was characterized by EDS mapping (Appendix A), HRTEM (Appendix A), TEM (Appendix A), and ICP-OES (Appendix A). The content of Pd was well kept after the five recycles indicated by the ICP-OES characterization (Appendix A). No agglomerated palladium particles were observed, as there had been no crystalline domains corresponding to Pd in the HRTEM (Appendix A), and the TEM mappings exhibited a uniform distribution of Pd (Appendix A). These results showed very well the reusable performance of the Cu@Pd/C catalysts.

## 4. Conclusions

In summary, the pulse oxidation was an effective method to regulate the oxidation degree of Cu in Cu/C catalysts, which could control the dispersion of Pd on Cu nanoparticles in the preparation of Cu@Pd/C catalysts by the galvanic reduction method. Surface Pd-rich Cu@Pd/C catalysts were allowed to form by the galvanic reduction method and had more superior catalytic performance in the hydrolysis of AB compared to the traditional method. The Cu(I) in the partially oxidized Cu/C did not react with Pd(II) and restricted the reduction location of Pd(II) as well as the aggregation of Pd(0) to form high-performance catalysts for the hydrolysis of AB. The TOF for the hydrolysis of AB reached 465 mol_H__2_ min^−1^ mol_Pd_^−1^ on Cu@Pd_0.5_/C-640. The catalysts had good stability in five continuous processes, which makes them promising candidates in the practical application of hydrogen evolution from ammonia borane. The manuscript also provides good inspiration for the preparation of high-performance noble metal-based catalysts.

## Figures and Tables

**Figure 1 nanomaterials-10-01850-f001:**
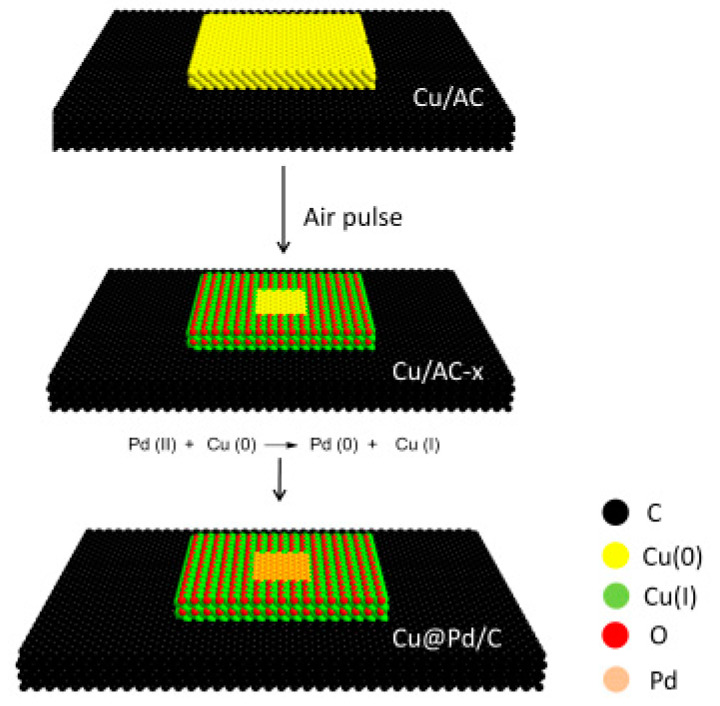
Procedure for the preparation of Cu@Pd/C.

**Figure 2 nanomaterials-10-01850-f002:**
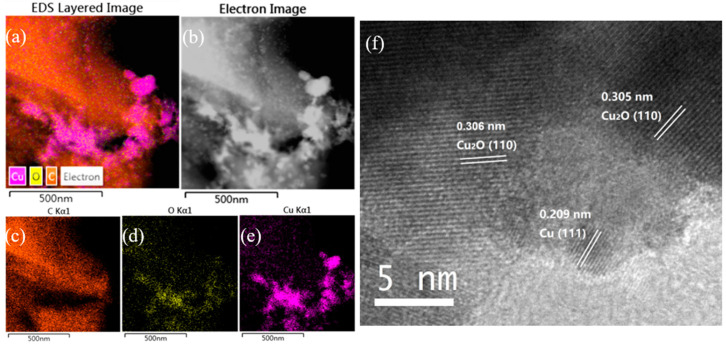
Energy-dispersive spectroscopy (EDS) mapping images (**a**–**e**) of Cu/C-320 and HRTEM (**f**) image of Cu/C-320.

**Figure 3 nanomaterials-10-01850-f003:**
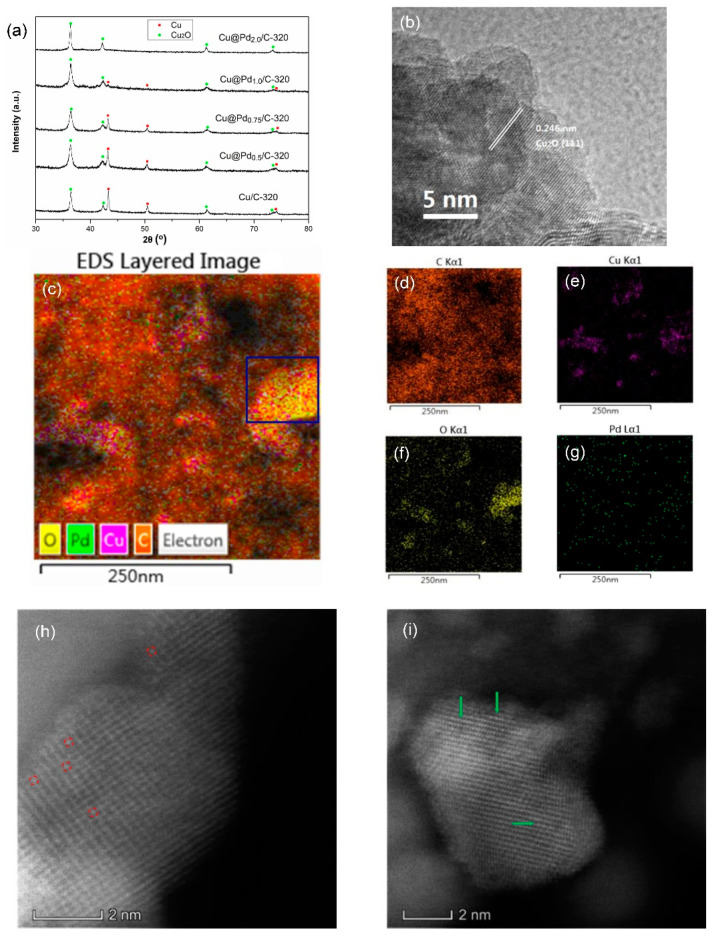
XRD patterns (**a**) of Cu@Pd/C-320, HRTEM (**b**), EDS mapping images (**c**–**g**), and high angular annular dark field scanning TEM (HAADF-STEM) images (**h**,**i**) of Cu@Pd_0.5_/C-320. The single Pd atoms (indicated by red circles) and Pd clusters (indicated by green arrows) could be seen in the HAADF-STEM images.

**Figure 4 nanomaterials-10-01850-f004:**
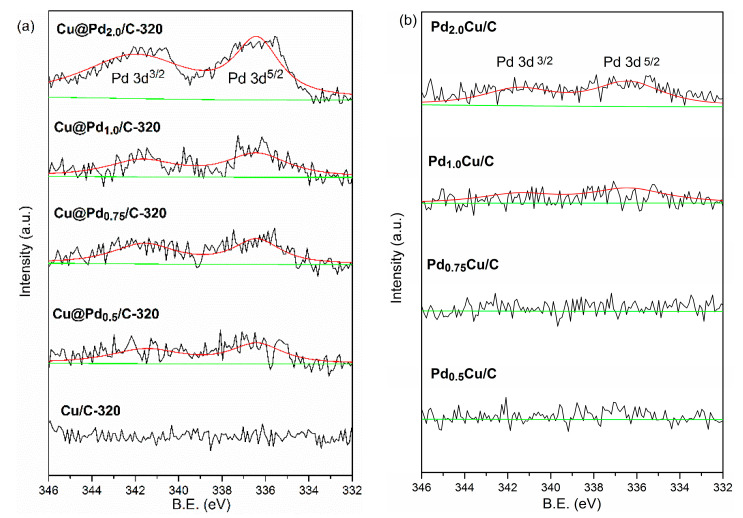
XPS spectra of Cu@Pd/C-320: Cu/C-320 (**a**), and PdCu/C (**b**).

**Figure 5 nanomaterials-10-01850-f005:**
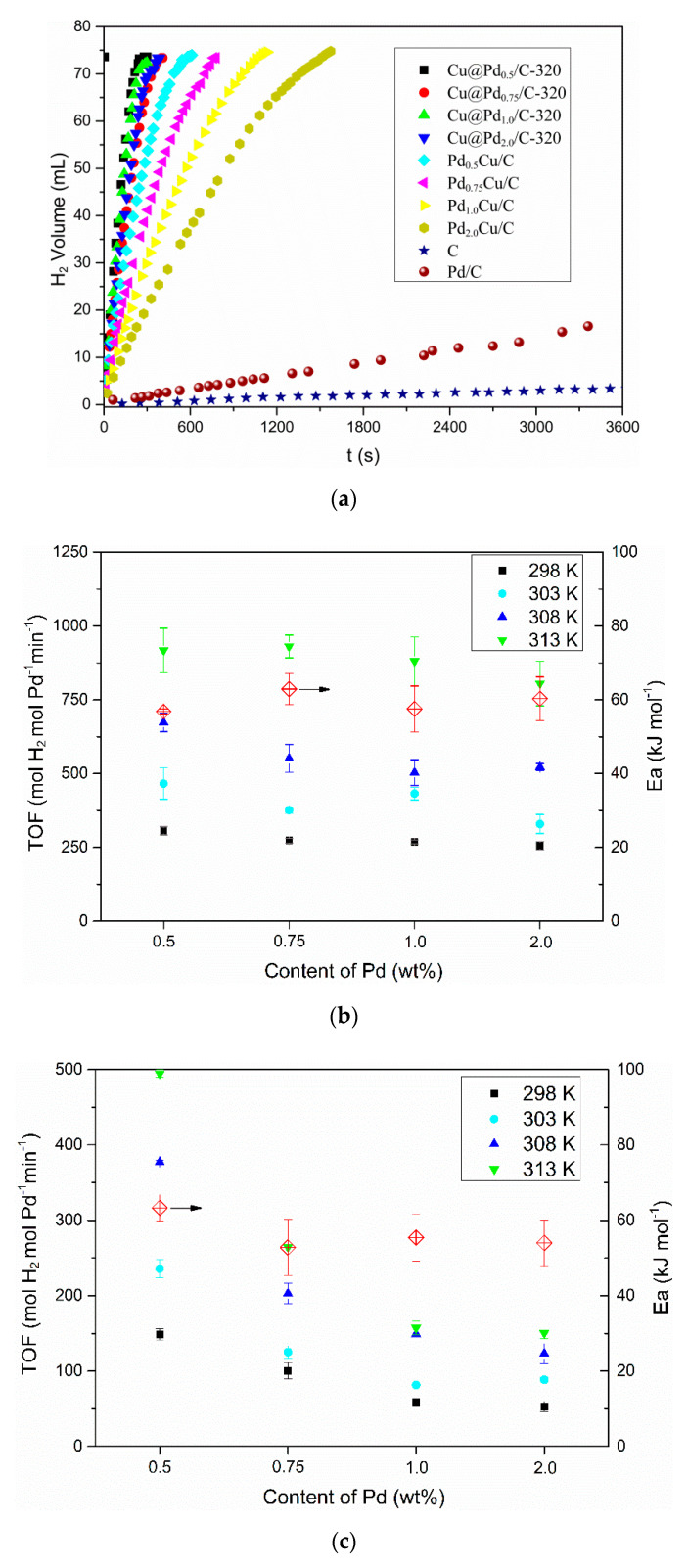
Plots of hydrogen evolution from ammonia borane vs. time over different catalysts at 298 K (**a**), TOFs on Cu@Pd/C-320 (**b**) and PdCu/C (**c**) at different temperatures and the corresponding Ea.

**Figure 6 nanomaterials-10-01850-f006:**
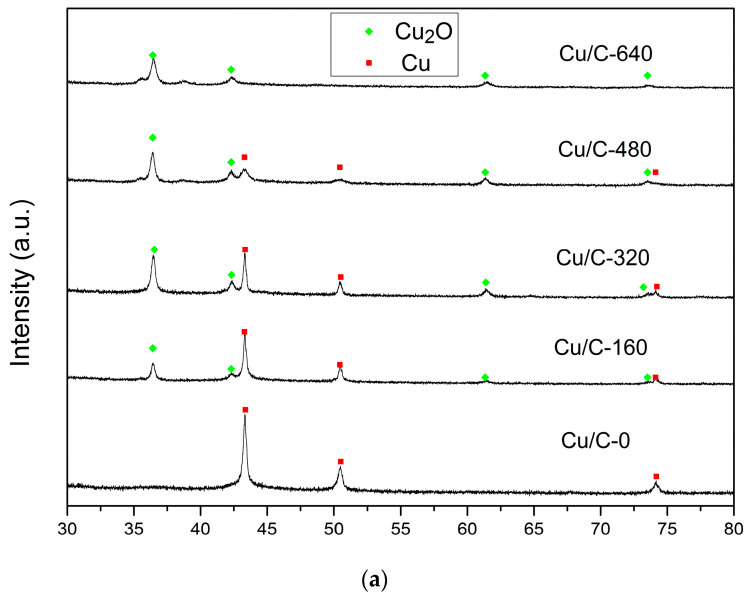
XRD patterns (**a**) XPS (**b**) and auger electron spectroscopy (AES) spectra (**c**) of Cu/C with different oxidation degrees.

**Figure 7 nanomaterials-10-01850-f007:**
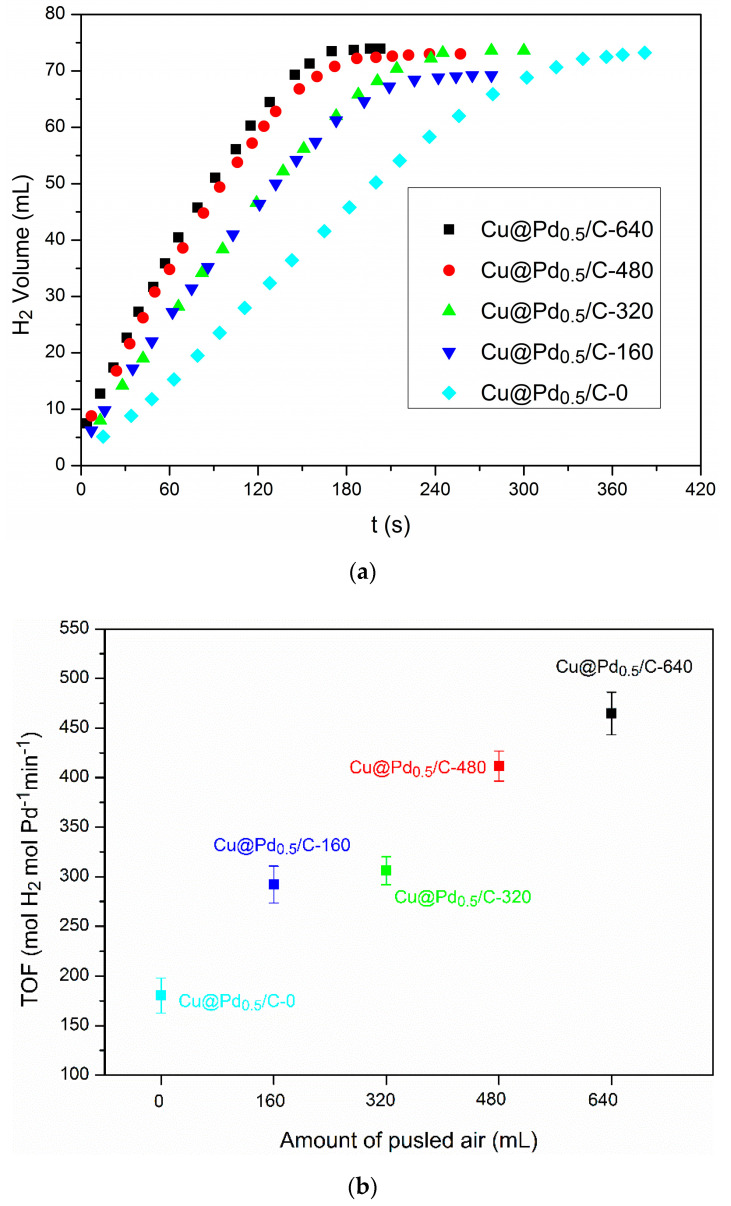
Plots of hydrogen evolution from ammonia borane vs. time over Cu@Pd/C prepared from Cu/C with different oxidation degrees (**a**) and the corresponding turnover frequencies (TOFs) (**b**).

**Figure 8 nanomaterials-10-01850-f008:**
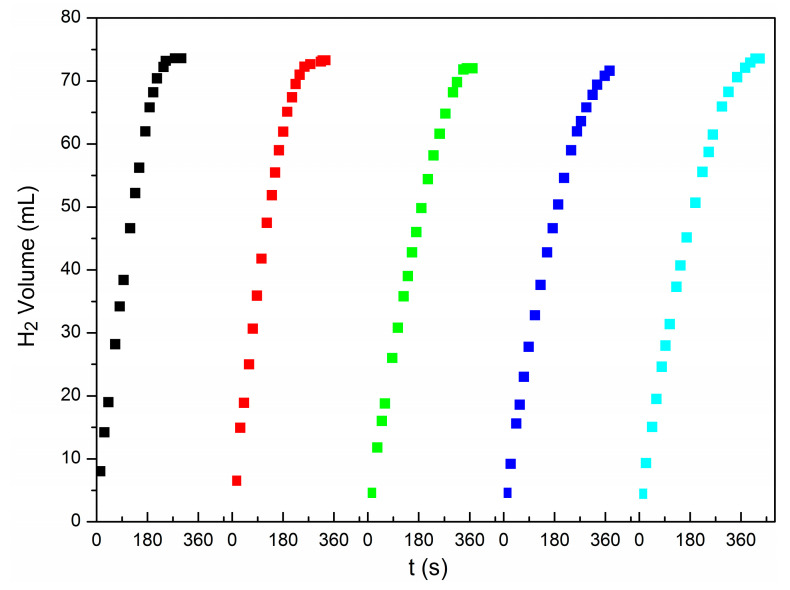
The stability test of Cu@Pd_0.5_/C-320.

**Table 1 nanomaterials-10-01850-t001:** Catalytic activity for the reported palladium catalysts in the hydrolysis of ammonia borane (AB).

Entry	Catalysts	Temperature (K)	TOF (mol_H2_ min^−1^ mol_Pd_^−1^)	*Ea* (kJ mol^−1^)	Ref.
1	Cu@Pd_0.5_/C-640	298	465	-	This work
2	Cu@Pd_0.5_/C-320	298	306	57	This work
3	Pd^0^/CoFe_2_O_4_	298	290	42	[38]
4	Pd(0)/SiO_2_–CoFe_2_O_4_	298	254	52	[33]
5	Pd_74_Ni_26_/MCN	rt	247	54	[39]
6	Pd@UiO-66	353	231	37	[40]
7	Pd/RCC3	303	176	-	[11]
8	Pd^0^/PDA–CoFe_2_O_4_	298	175	65	[38]
9	Pd/IPCNs	298	113	29	[29]
10	Pd/AC	303	40	68	[32]
11	Pd^0^/CeFe	298	29	-	[28]
12	Pd(0)/g-C_3_N_4_-CS	303	28	35	[15]
13	Pd/CGP-GO-Fe_3_O_4_	303	27	37	[36]
14	Pd(0)/GO-ILCS	303	26	38	[14]
15	Pd(0)/CS-Fe_3_O_4_	303	15	36	[37]

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
