# Peer review of "Cu@Pd/C with Controllable Pd Dispersion as a Highly Efficient Catalyst for Hydrogen Evolution from Ammonia Borane"

_nanomaterials, 2020, doi:10.3390/nano10091850_

Round 1
Reviewer 1 Report
Author reported Cu@Pd/C (via galvanic reduction method) showed good Hydrogen evolution over PdCu/C via impregnation method. The results are quite interesting. It will be great if author provide a table with best materials for hydrogen evolution from ammonia Borane and see where this catalyst stands. I also some ambiguity of the XPS spectral analysis of Pd in figure 4. I would prefer to see a clear peak but they are almost not perfect.
Reviewer 2 Report
This is an interesting study, which enriches the scientific knowledge in terms of catalytic sites size control. Although some of the results are rather expected (increased oxidation of Cu over amount of O2 used) the study is sound and the data might be useful for researchers in the same area or even for extending the idea to other systems. I propose publishing this work, but some significant changes are required:
- The abstract is rather difficult to follow at the first reading. One must go to the content in order to understand the meaning of the different names used. I suggest presenting an introductory sentence explaining the symbolism.
- Reproducibility of the testing is completely missing. We have no clues about the systematic error of the testing. Without this kind of information the conclusions, although seeming fully supported by the observed differences, they cannot be statistically validated.
- In Figures 5 and 7, the authors use lines to link the points. Since there is no mathematical model, the use of lines is misleading and pointless. The differentiation of the points using geometric symbols and colours is sufficient.
- Although the instrumental analysis of the catalytic formulation is good and informative, there are no similar analyses on used catalysts. Why?
Round 2
Reviewer 2 Report
All questions and suggestions have been taken into account satisfactorily. I recomment publishing this manuscript